# Intelligent Energy Management Algorithms for EV-charging Scheduling with Consideration of Multiple EV Charging Modes

**Tian Mao [1], Xin Zhang [2,\*] and Baorong Zhou [1]**

[1]   State Key Laboratory of HVDC, Electric Power Research Institute, China Southern Power Grid, Guangzhou 510663, China; maotian@csg.cn (T.M.); zhoubr@csg.cn (B.Z.)
[2]   School of Electrical and Electronic Engineering, Nanyang Technological University, 50 Nanyang Avenue, Singapore 639798, Singapore
\*   Correspondence: jackzhang@ntu.edu.sg; Tel.: +65-6790-5419

**Abstract:** Electric vehicles (EVs) are now attracting increasing interest from both industries and countries as an environmentally friendly and energy efficient mode of travel. Therefore, the EV charging and/or discharging issue has become an important challenge and research topic in power systems in recent years. An advanced and economic EV charging process, however, should employ smart scheduling, which depends on effective and robust algorithms. To that end, a comprehensive intelligent scatter search (ISS) algorithm within the frame of a basic scatter search has been designed with both unidirectional and bidirectional charging considered. The ISS structure also supports both a flexible and constant charging power rate by respectively employing filter-SQP (sequential quadratic programming) and mixed-integer SQP as local solvers with module control. The detailed design of ISS is presented and the objectives of smoothing the daily load profile and minimizing the charging cost have been tested. Compared with methods based on GS (global search), GA (genetic algorithm), and PSO (particle swarm optimization), the outcome-verified ISS can produce attractive results with a significantly short computational time. Moreover, to handle a large scale EV charging scenario, a hybrid method comprised of a GA and ISS approach has been further developed. Simulation results also verified its prominent performance, plus superbly low computational time.

**Keywords:** charging/discharging; electric vehicle; energy management; genetic algorithm; intelligent scatter search

## 1. Introduction

The prominent advantages of electric vehicles (EVs) lie in their high energy transfer efficiency and lack of carbon dioxide ($CO_2$) and other air pollutant emissions. The transition to replace petrol vehicles by EVs, however, is never an easy task. From the aspect of power systems, conventional power distribution networks are typically designed for non-mobile loads, meaning that current power system infrastructures may not be resilient enough to accommodate the integration of EVs [1]. For instance, the daily load peak could be amplified by the aggregated load of unregulated EVs [2,3], which is certainly a threat for power grids. Other potential adverse impacts including voltage drops, frequency oscillations, network congestions, power imbalances, etc., as well concern for power system operators [4,5]. In addition, the implementation of V2G (Vehicle-to-grid) also affects the dynamics and performance of the entire power grid [6,7].

Reinforcing power infrastructures is a typical approach for addressing the above challenges, yet such a method appears to be inefficient and environmentally unfriendly. Meanwhile, it is also impossible to upgrade current power facilities for the accommodation of EV penetration, due to the

expensive investment. As a more economical and attractive alternative, smart scheduling strategies can be employed to postpone the construction of power infrastructures to support EV charging demand, through which the goal of wise load regulation as well as economic benefits can be reached [8–10]. Nevertheless, owing to the differences in operational characteristics between EV and the power system, smart EV scheduling becomes a rather complex task that must rely on an efficient and robust charging algorithm [11]. In general, the operation of EVs should take into account, but not be limited to, a charging and/or discharging power rate limit, initial state of charge (SOC), customer travelling habits, final energy demand, and battery capacity. On the other hand, the operation of a power system is constrained from the limitations of generation units, network structure, transformer capacities, voltage and frequency requirements, etc. In addition, power grids themselves are dynamic networks with unpredictable internal complexities and volatilities, indicating that real-time charging schemes are more practical and superior than merely day-ahead negotiation. As of now, several algorithms using combinations of different solving techniques have also been developed to solve real-time EV scheduling problems [12,13]. With all the above considerations, the demand for powerful algorithms to perform intelligent EV charging scheduling is apparent [14].

As of now, several approaches or algorithms attempting to regulate EV charging have been designed and reported. Typically, the constraint functions related to EV scheduling are usually, or particularly, linear. For example, the EV charging/discharging power should be bounded within the lower and upper limits, whereas the aggregated load must be constrained by the power supply. In the meantime, the objective functions (e.g., cost minimization, load smoothing) can be also formed or reformulated as convex functions. Hence, the EV scheduling issue can be modeled as a convex optimization problem and solved through CVX [15], GAMS [16] or CPLEX [17]. However, these tools come with license concerns, i.e., one must pay for their license to legally use these tools. On the other hand, many conventional methods, such as the alternating direction method of multipliers [18] and tree-based dynamic programming [19], have also been employed to solve the intractable issues of EVs.With multi-constraints and multi-objectives, this kind of approach often dividesthe problem into several sub-problems, thereafter solving them over each stage. When the number of EVs is increased, however, these methods can get stuck, due to "curse of dimensionality" [20].This can limit their usage in large-scale EV charging.Furthermore, some heuristic algorithms inspired from nature process such as a particle swarm optimization (PSO) [21], CRO (chemical reaction optimization) [22], and genetic algorithm (GA) [23,24] have also been utilized. These artificial intelligent algorithms, as verified, can outperform conventional methods in terms of computation overhead, with promising results. For example, the PSO approach in [21] for implementing EVs demand response is almost 2700 times faster than the MINLP (mixed integer nonlinear programming) technique under different scenarios. The deficiency of such methods is that they are intrinsically stochastic and can stagnate prematurely into local optima.

Although many algorithms have been utilized for scheduling EV charging, the emerging approaches are limited in a sense that the methods either only consider EV charging by assigning its states at an arbitrary value from 0 to a specified charging power limit (flexible power rate charging) [10,21], can just manage the EV charging pattern through adjusting the charging status (constant power rate charging) [25], or focus merely on unidirectional charging. Nevertheless, in the long run, both V2G and a flexible charging rate can occur simultaneously with unidirectional charging and a constant power charging rate, due to the fact that the future smart grid will accommodate various applications and services [26]. All the above issues indicate that a generic algorithm is desirable to meet the current EV charging demand and also support future bidirectional charging.

Hence, the aim of this work is to design a comprehensive and universal algorithm that can fit multiple charging modes and diverse charging rate scenarios for distribution-side management. With such complex considerations, an intelligent scatter search (ISS) algorithm framework has been designed. In principal, this novel method is essentially a hybrid SS (scatter search) framework, integrated with sequential quadratic programming (SQP), based local solvers. SS in nature is a population-based

evolutionary approach [27]. Instead of depending on massive random components and traditional evolution techniques, like mutation and crossover, SS only concentrates on a small solution set. Besides, SS is attractive for its well-designed flexibility, enabling various local search methods to complete different optimization tasks. In the design, SQP techniques are chosen as the local solvers to cope with different charging power rates, since they are efficient in solving nonlinear constrained problems. More specifically, a trust-region SQP method based on a filter technique (filter-SQP) [28] is adopted for solving the flexible power rate charging, while the MISQP (Mixed-integer SQP) technique evolved in [29] is in charge of constant power rate charging.

The main contributions of this work lie in: the general SS framework is redefined and adapted to be a new algorithm framework; for the scheduling of EV charging, two algorithms are proposed and applied, i.e., an ISS framework to deal with single EV charging, and a GA-ISS method comprised of GA theory and the proposed ISS approach for massive EV charging; and the proposed algorithms can support both V2G and G2V (grid-to-vehicle) for different power rate configurations.

For comprehensive comparison, various algorithms are compared in the simulation part, specifically:

(1)　For the ISS framework, GA, PSO, and global search (GS) are compared;
(2)　For the GA-ISS framework, three methods are compared, including: a global control that calculates all the variables and constraints through CVX; DCM (dumbing control method) that charges all EVs as soon as they are plugged in; and a GA-PSO hybrid method.

The following paper is organized as follows: the modeling of the constraints and objective functions for the optimization of EV charging is described in Section 2, the detail designs of the proposed algorithms are presented in Section 3, simulation results to demonstrate and verify the effectiveness of the proposed ISS and GA-ISS are described in Section 4, and finally, a conclusion is given in Section 5.

## 2. Modelling for the Optimization of EV Charging

### 2.1. Assumptions

Normally, an EV completes its charging within a certain time window, rather than a whole day. Within this time frame, the charging status (either charging, discharging, or idling) together with the charging power rate, are required to be determined for each timeslot. The following assumptions have been made for the algorithm development:

(1)　All participated EVs are with the ideal charging/discharging efficiency of 100%.
(2)　The charging price and discharging tariffs follow the spot prices of basic load at each timeslot.
(3)　The time period of study starts from 8:00AM to 8:00AM of following day with 15 min per interval. Therefore, the time window is divided into 96 timeslots.

### 2.2. Definition of Charging Modes

Since both unidirectional and bidirectional charging, combined with either flexible or constant charging power rate, are considered, to better elaborate the design, four charging modes are herein defined:

(1)　CD-F mode: charging/discharging with a flexible charging rate,
(2)　CD-C mode: charging/discharging with a constant charging rate,
(3)　C-F mode: charging only with a flexible charging rate,
(4)　C-C mode: charging only with a constant charging rate.

The detailed comparisons for the different charging modes are listed in Table 1. Based on the type of variables, the charging problems can be transformed into NLP (nonlinear programming) or MINLP sub-problems, which will be introduced later.

**Table 1.** Comparisons for the four charging modes.

| Charging Modes | Charging Rate Range (Normalized Value) | Variables Type | Sub-Problem |
|:---:|:---:|:---:|:---:|
| CD-F | [−1,1] | Continuous | NLP |
| CD-C | [−1,1] | Discrete | MINLP |
| C-F | [0,1] | Continuous | NLP |
| C-C | [0,1] | Discrete | MINLP |

*2.3. Constraints*

With practical considerations, EV scheduling is subjected to its charging power limit, battery capacity, and customer demand, while the total load is constrained by the supply capacity of power grid.

(1)  Charging power limit

Generally, the value of $P_{EV,j}^{\max}$ is equal to the battery nominal charging power rate, $P_{EV,j}^{\min}$ is set to be zero for charging only cases while set to be same as $P_{EV,j}^{\max}$ when V2G is permitted, i.e.,

$$- P_{EV,j}^{\min} \leq P_{EV,j}^{i} \leq P_{EV,j}^{\max} \qquad \forall i \in \text{ST}, \forall j \in \text{SE}. \tag{1}$$

(2)  Restriction of the battery capacity

For the sake of prolonging the lifetime of EV batteries, $Soc_{j,i}$ is often limited between $Soc_{j,\min}$ and $Soc_{j,\max}$. Take commonly used Li-ion batteries as an example, the range $Soc_{j,\min}$ and $Soc_{j,\max}$ are normally set to be 20% and 90% of their nominal state-of-charge (SOC) $E_{RC,j}$, respectively [30]. Therefore, the following constraints are considered:

$$Soc_{j,\min} \leq Soc_{j,i} \leq Soc_{j,\max} \leq E_{RC,j}, \tag{2}$$

$$Soc_{j,i} = Soc_{j,start} + \frac{1}{C_{EV,j}} \sum_{i=start}^{current} P_{EV,j}^{i}. \tag{3}$$

(3)  SOC expectation level for journey requirement

To fulfill the journey requirement, the final energy state $Soc_{j,end}$ of the battery must reach $E_{exp,j}$, i.e.,

$$Soc_{j,\text{end}} = Soc_{j,start} + \frac{1}{C_{EV,j}} \sum_{i=start}^{end} P_{EV,j}^{i} = E_{\exp,j}. \tag{4}$$

(4)  Capacity constraint of power grid

To ensure normal operation of the power grid, the total load cannot exceed the maximum capacity of generation units, i.e.,

$$P_{total}^{i} \leq P_{g}^{i}, \tag{5}$$

$$P_{total}^{i} = P_{ld}^{i} + \sum_{j \in SE} P_{EV,j}^{i}. \tag{6}$$

*2.4. Objective Functions*

In terms of optimization objectives, the emerging applications for EV regulation normally consider: (i) load curve flattening, (ii) energy cost minimization, (iii) running cost reduction for the power system, (iv) benefit maximization for aggregators, (v) reduction of $CO_2$ emissions, and (vi) combinations of the above. The goal of multiple purpose optimizations can be set by utilizing weight factors, based on the

emphases of different entities. For exemplification's sake, only the first two objectives are considered to verify the proposed scheduling algorithm.

(1)  Load curve flattening

This optimization objective aims to minimize the power load fluctuation, and the standard deviation of load profile can be utilized for this purpose:

$$\text{Minimize} \quad F_{obj,1} = \left[ \frac{1}{n-1} \sum_{i \in ST} (P_{total}^i - P_{total}^{Ave})^2 \right]^{1/2}, \tag{7}$$

where $P_{total}^{Ave}$ denotes the daily load mean value and calculated as:

$$P_{total}^{Ave} = \frac{1}{n} \sum_{i \in ST} P_{total}^i. \tag{8}$$

(2)  Energy cost minimization

The objective function to minimize power energy cost is given as:

$$\text{Minimize} \quad F_{obj,2} = \sum_{i \in ST} P_{total}^i C_e^i \tag{9}$$

The two mostly studied pricing strategies, namely time-of-use (TOU) price and real-time price, will be used for the simulation to determine the electricity price. The former usually sets the price $C_{TOU}^i$ for different timeslots, while the latter determines an electricity tariff strictly according to time and load demand. For convenient purposes, the linear-price (LP) given in Equation (10) is adopted to model the dynamic real-time price. By incorporating Equation (10) into Equation (9), the final form of the function for cost minimization with the LP price is obtained and given in Equation (11), which is in a quadratic polynomial form.

$$C_{LP}^i = \psi P_{total}^i + \gamma \tag{10}$$

$$\begin{aligned} \text{Minimize} \quad F_{obj,2} = \sum_{i \in ST} P_{total}^i C_{LP}^i &= \sum_{i \in ST} P_{total}^i (\psi P_{total}^i + \gamma) \\ &= \sum_{i \in ST} (\psi {P_{total}^i}^2 + \gamma P_{total}^i) \end{aligned} \tag{11}$$

As can be seen, the optimization objectives in Equations (7), (9), and (11) are in three different forms, which will be separately simulated.

*2.5. Transform the EV Charging into NLP/MINLP Problem*

As previously introduced, the constraints considered for EV charging are nonlinear. Therefore, the EV charging cases are transformed into NLP or MINLP optimization problems [16,31], which consist of an objective function, some box bound constraints, several equal, and unequal constrained functions. Generally, the NLP problem has the form:

$$\begin{aligned} &\text{minimize } F_{obj}(x), \\ \text{subject to}: \quad &H_{eq}(x) = 0 \\ &G_{ieq}(x) \leq 0 \\ &l_b \leq x \leq u_b \end{aligned} \tag{12}$$

where $x$ represents the controllable vectors, $F_{obj}(x)$ denotes the objective function, $H_{eq}(x)$ and $G_{ieq}(x)$ represent the equality and inequality constraints, and $l_b$ and $u_b$ are the lower and upper box limits respectively, all with compatible dimensions.

In this work, the issue of flexible charging rate (CD-F and C-F mode) is therefore transformed into an NLP problem. For the condition of constant charging rate (CD-C and C-C mode), only the charging status of EVs needs to be settled, and the Equation (1) thus evolves into Equation (13), and the optimization model turns into MINLP.

$$
\begin{aligned}
P^i_{EV,j} &= P^i_{EV,j} S^i_{EV,j} \quad \forall i \in \text{ST}, \forall j \in \text{SE} \\
S^i_{EV,j} &= \begin{cases} 1, & \text{charging} \\ -1, & \text{discharging, if permitted} \\ 0, & \text{otherwise} \end{cases}
\end{aligned}
\tag{13}
$$

## 3. Proposed ISS and GA-ISS Algorithm Frameworks

There are two algorithms proposed in this work: ISS for single EV charging and GA-ISS for massive EV charging. The proposed ISS algorithm is based on a basic scatter search (SS) framework with advanced local solvers. Therefore, to better understand the proposed algorithms, the principle of SS and the utilized local search solvers will first be described, followed by the ISS framework, and the hybrid method GA-ISS handling massive EVs.

### 3.1. Basics of Scatter Search (SS)

SS is a novel evolutionary-based method, since it normally avoids too many random components (e.g., mutation or crossover operators), which other methods, such as GA, rely on [27]. It has been reported that SS has been successfully applied in a variety of continuous and discrete optimization problems [27,32–34]. One prominent feature of a scatter search lies in its flexible structure, where a variety of ways and degrees of sophistication can be deployed for its elements [27]. Moreover, instead of operating on massive populations, SS concentrates only on a small set denoted herein as *RefSet* [32–34]. It obtains an optimum through balancing diversification (robustness) and intensification (efficiency) for *RefSet* systematically. The whole process of basic SS is shown in Figure 1.

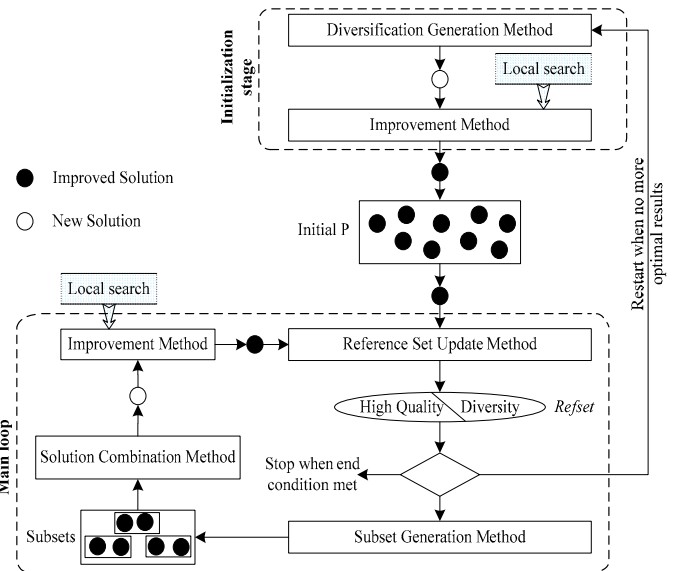

**Figure 1.** Schematic of a standard scatter search (SS) template.

In general, SS contains five strategies in total [27]: (i) a diversification generation method is used for generating an original population *P*, (ii) an improvement method wherein local search procedures can be embedded to examine the trial individuals and acquire high quality solutions, (iii) a reference set update method to initialize and maintain *RefSet*, normally resulting in a high quality subset *RefSet*₁ with best fitness value and a diversified subset *RefSet*₂ with optimal diverse

value ($RefSet = RefSet_1 + RefSet_2$), (iv) a subset generation method operates on $RefSet$, via which subsets are created and then utilized for producing combined vectors, and (v) a solution combination method for computing new testing vectors (one or more) from the results obtained from the subset generation method.

Optionally, a restart phase can be scheduled to rebuild the solutions. However, the SS framework described above is just a generic structure and all the five methods will be clearly defined in the latter designed ISS method.

### 3.2. The Utilized Local Search Solvers

As previously introduced, different SQP techniques, including filter-SQP and MISQP, are chosen as local solvers to complete different tasks. The process of the filter-SQP, as shown in Figure 2, essentially follows the procedure described in [28]. At each iteration, the sub-problem is replaced and solved as the trust-region sub-problem $TRQP(x_k, \Delta_k)$, with $\Delta_k$ denoting the trust-region radius. For trust-region methods, however, a common problem lies in the difficulty of tuning penalty coefficients. A filter technique is hence incorporated to determine whether the solution outperforms the previous one and also guarantee the acceptability. The acceptance of the basic filter is defined by comparing the violation function value $h(x_k)$ and fitness value $f(x_k)$:

$$h(x_k) \leq \beta h_{\mathcal{F}} \ \ or \ \ f(x_k) \leq f_{\mathcal{F}} - \lambda h_{\mathcal{F}} \ \text{for all} \ (h_{\mathcal{F}}, f_{\mathcal{F}}) \in \mathcal{F}, \tag{14}$$

where $0 < \lambda < \beta < 1$, $\mathcal{F}$ represents the filter set, $h_{\mathcal{F}}$ and $f_{\mathcal{F}}$ are the respective violation and fitness values of the element in the set. Additionally, with the aim of improving feasibility and optimality, a novel non-monotone technique for setting the filter criteria is also implemented [28]. In essence, it sets the filter by dividing the trail step $d_k$ into a quasi-normal part $d_k^n$ and a tangential part $d_k^t$.

---

**Filter-SQP optimization process**

**Input:** Parameters of the problem

**Output:** The best solution

　Step 1) Initialization for the variables, initial region radius $\Delta_0 \geq \Delta_{\min} > 0$, Hessian matrix, $\mathcal{F} = \{(h_o, f_o)\}$;

　Step 2) Solve the $TRQP(x_k, \Delta_k)$, stop if terminal condition is met

　Step 3) Calculate $d_k^n$ and $d_k^t$, obtain $d_k = d_k^n + d_k^t$;

　Step 4) Judge whether $x_k + d_k$ is acceptable to $\mathcal{F}$. If yes, execute step 5, otherwise enter into step 6;

　Step 5) When $d_k$ is unacceptable, enter into step 7, else execute step 6;

　Step 6) Adjust $\Delta_k$, back to step 3;

　Step 7) Update $x_{k+1} = x_k + d_k$ and $\mathcal{F}$, update $\Delta_{k+1}$, Hessian matrix. Loop back to step 2 (for next iteration).

**Figure 2.** Pseudo code of filter- sequential quadratic programming (SQP).

For CD-C and C-C charging modes, the problem turns out to be far more complex, since no explicit available criterion can be utilized to approach the optimal solution, nor can the optimal positions of the integers be effectively captured from relevant corresponding continuous solutions. The MISQP technique in [29] is therefore employed for solving the MINLP model in the proposed framework. Three key techniques are employed to achieve the tasks and they are: (i) a trust region technique with second-order amendments to stabilize the algorithm, (ii) a quasi-Newton formula to update the Hessian matrix, and (iii) a branch-and-bound skill to handle the sub-problems.

*3.3. The Proposed ISS Algorithm for Single EV Charging*

The implementation of the proposed ISS algorithm follows the generic framework of SS as shown in Figure 1. Several techniques have been introduced to enhance the overall performance.

(1)　Initialization and restart mechanism

A diversification generation method is implemented with Latin hypercube uniform sampling [33]. The initial results are then filtered as the original population $P$, of which half ($RefSet_1$) are top-ranking solutions with better fitness values, while another half ($RefSet_2$) are arbitrarily selected from the rest of the diverse vectors. On top of this, a restart mechanism that aims to escape from local minima is also designed when improvement cannot be obtained after the subset updating step. For this process, while leaving the subset $RefSet_1$ unchanged, additional solutions are first created with the diversification generation method, and the subset $RefSet_2$ is reconstructed in the manner of subsequent updating method.

(2)　Improvement procedure

The improvement procedure embedded with local search solvers is utilized to improve the results obtained from the diversification or the combination methods. Since different objectives and price types are considered for both NLP and MINLP problems, an intelligent module is designed to conduct the optimizations, as shown in Figure 3. The data will be categorized into different types, including: power rate type, charging type, price type selection, and objective selection, with which the charging power rate, charging type, price category and objective function would be determined respectively. All this information is then collected by the joint units where different local solvers are allocated.

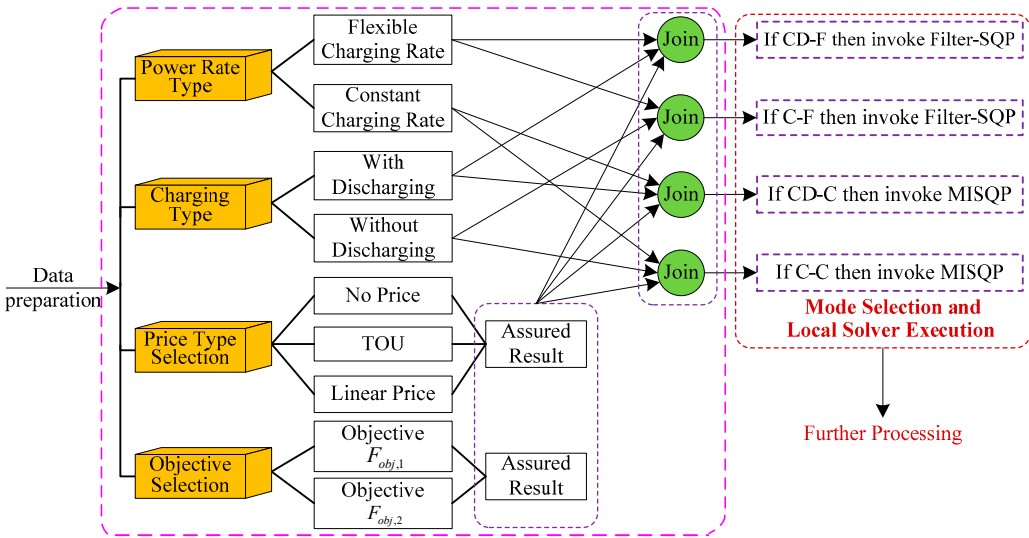

**Figure 3.** The intelligent module for executing the local solvers of the intelligent scatter search (ISS).

(3)　Population updating

For the update of the solutions, a new solution can only enter into $RefSet_1$ upon the following criteria: (i) the individual occupies a fitness value superior to the worst one in $RefSet_1$, or (ii) the individual has a diverse value larger than the worst one in $RefSet_2$. The diverse value is assessed according to the Euclidean distance $d(x,z)$ between the candidate and the individuals in $RefSet_1$. The new solution that maximize $d_{min}(x)$ will be selected to update $RefSet_2$:

$$d_{min}(x) = \min_{z \in RefSet_1} \{ d(x,z) \}. \tag{15}$$

(4)　Subset generation and combination

A common strategy considering all pairs of the individual solutions is adopted to generate the basis for creating combined solutions. Instead of using simple linear combination, an improved hyper-rectangle based skill proposed in [34] has been chosen for this task. For each pairwise member, a bias of their relative position is utilized to define the hyper-rectangles for leading the solutions to approach "better" ones and move away from "bad" ones. This method can help to extend the search region and also expand the exploring directions without an additional memory term. Meanwhile, the "go-beyond" strategy from the same paper is also employed to enhance the intensification. The principle of this technique is to search the potential directions of the best generated individual and its parents continuously, in order to produce new individuals within two generations. Since the search area in this process is extended, the strategy promotes a diversity of solutions and accelerates convergence.

To further confine each element $x_k^r$ of the solution vector within the correct search region, the repair strategy given below is employed:

$$
\begin{aligned}
&\text{if } \ x_k^r > u_b^r, \quad \text{then } x_k^r = u_b^r \\
&\text{if } \ x_k^r < l_b^r, \quad \text{then } x_k^r = l_b^r
\end{aligned}
\tag{16}
$$

where $l_b^r$ and $u_b^r$ are respective lower and upper limits.

### 3.4. The Proposed GA-ISS Method for Massive EV Charging

When massive EV charging is simultaneously considered, the computation burden is enormous. Neither traditional deterministic techniques nor intelligent meta-heuristics can easily produce the optimized result, since: (i) the CPU may run out of memory due to large data demand, and (ii) long computation time is expected because of its complexity. For instance, a whole day is required to solve a single EV charging case that comes with 96 variables. Accordingly, there are 96 box constraints for Equation (1); 1 equality constraint for Equation (4), and 96 × (2 + 1) inequality constraints for Equations (2) and (5). Therefore, it is desirable to have an efficient algorithm to deliver accurate solutions.

To handle large-scale EV charging, a hybrid method integrating GA and ISS is proposed, as shown in Figure 4.

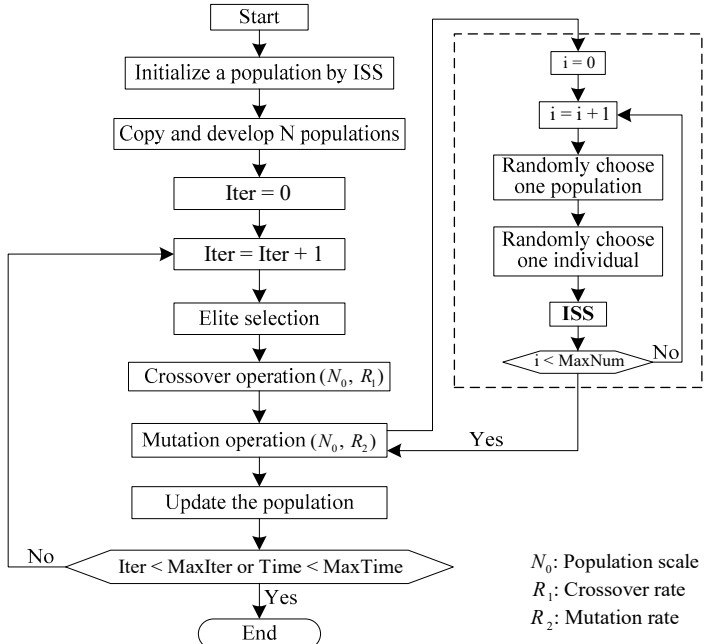

**Figure 4.** A genetic algorithm (GA)-ISS based hybrid method.

In Figure 4, $N_0$, $R_1$ and $R_2$ denote the population scale, crossover, and mutation rate, respectively. The proposed method (GA-ISS) expects to benefit from the gist of both algorithms, for which GA processes present a strong global search ability and robustness, while the ISS structure is prominent in searching capability and quick convergence. During the procedure, a single EV state is determined by ISS, whilst the entire fleet co-evolves to the optimum and relies on the mutation and crossover procedures from GA.

To speed up the process, the initialization of population is first processed by ISS and copied into the $N_0$ populations, which will then be optimized in the main loop. The mutation operation takes the speed and optimization advantage of the ISS, through which individuals from the previous population are refined. As previous individuals are utilized as the starting points of the ISS, the time spent for refining the individuals will be substantially reduced, since the old ones have already been optimized to approach their optimal positions.

## 4. Simulation and Results

### 4.1. Parameter Setting

A case of overnight charging in a typical residential area where EV charging occurs during the night is considered in the simulations. The TOU price modeled from [35] is set on a rolling hourly basis, and the price curve as well as the tested basic daily load profile is shown in Figure 5. The LP rate coefficients are chosen to be $\psi = 2 \times 10^{-4}$ and $\gamma = 0.22$. In addition, the maximum generation capacity is limited to be 20% above the daily peak load.

A single EV is used to illustrate the effectiveness of the ISS. The battery parameters and charging behavior are given in Table 2. It is also assumed that the minimum allowed SOC of the battery is 20%.

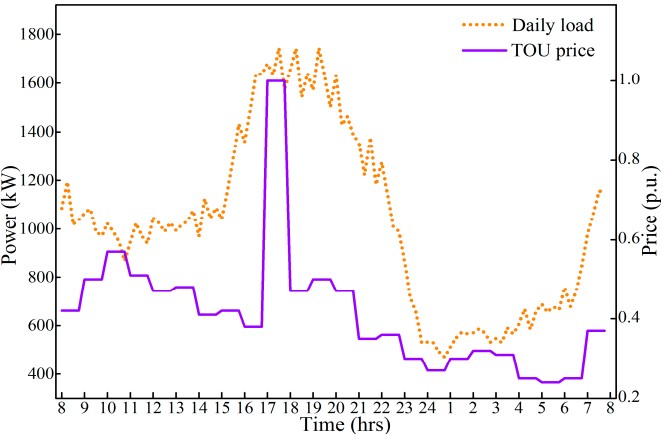

**Figure 5.** Data profile for simulation.

**Table 2.** Parameters for the single electric vehicle (EV).

| Capacity | Initial SOC | Rated Charging Power | Start Time | End Time | Desired SOC |
|---|---|---|---|---|---|
| 24 (kWh) | 58.59% | 3.3 kW | 21:15 | 6:30 | 90% |

To test the GA-ISS method, an EV fleet with 100 vehicles is simulated. As EVs arrive and depart at the charging points in a random manner, the arriving time, departure time, rated capacity, rated plugging power, and the initial SOC of the test fleet are defined based on truncated normal distribution [36]. In Table 3, the standard deviation (Std.), minimum, maximum, and mean values of the EV fleet are listed. It should be noted that the parameters of EVs in this work are just adopted for demonstration purposes, and there is no restriction on the power and capacity of EV batteries for the proposed algorithms.

Table 3. Parameters for the fleet with 100 EVs.

| Parameter | Minimum | Maximum | Mean | St. Dev |
|---|---|---|---|---|
| Arriving Time (h) | 18:00 | 22:00 | 20:00 | 1:30 |
| Departure Time (h) | 5:45 | 7:45 | 7:00 | 0:45 |
| SOC (%) | 20 | 90 | 50 | 20 |
| Capacity (kWh) | 10 | 30 | 18 | 6.93 |
| Plugging power (kW) | 2 | 10 | 3.54 | 1.48 |

*4.2. Simulation Results*

The algorithm is coded on the MATLAB Platform (R2012a) and executed on a PC with 2.66-GHz Quad CPU (Intel Core 2), 4GB RAM and Windows 7 64-bit system. The performance of the ISS is first tested with the single EV, followed by the testing for a GA-ISS with an EV fleet.

(1)  Single EV charging

For a single EV case, the algorithm can help the user determine the charging schedule to reduce the charging cost, or to better cooperate with the system operator to smooth the load profile. In order to evaluate the performance of ISS:

(a)  Two other commonly used heuristic algorithms, GA [37] and PSO [38], have been tested for flexible charging rate modes. And the GS method is utilized to produce the best reference result, since it explores all possible solutions [39].

(b)  Since a GS is unable to solve MINLP problems, only a GA and PSO [40] have been tested for constant charging rate.

(c)  The population number of the ISS is set to 30, while its iteration number of local solvers is limited to 10. The population member of the GA and PSO are set to 100. In addition, the total iteration numbers for all the above methods are set to 100.

Three different objective functions are used for the evaluation: (i) OF1—load curve flattening, refer to Equation (7), (ii) OF2—cost minimization with TOU price, refer to Equation (9), and (iii) OF3—cost minimization with LP price, refer to Equation (11).

For each condition, 50 simulations have been carried out, and Table 4 gives the mean values of the simulation results for all methods for comparison. It can be seen that the mean values for all the methods under different charging scenarios appear to be similar, however, an ISS can give slightly better performances compared with the heuristic GA and PSO algorithms for all three objective functions.

Table 4. Mean values of the methods for single EV charging.

| Objectives | Type | GS | GA | PSO | Proposed ISS |
|---|---|---|---|---|---|
| OF1<br>(standard deviation, kW) | CD-F | 368.19 | 368.91 | 368.30 | 368.21 |
| | CD-C | N/A | 368.40 | 368.40 | 368.36 |
| | C-F | 368.71 | 368.96 | 368.77 | 368.74 |
| | C-C | N/A | 369.00 | 368.80 | 368.80 |
| OF2<br>(normalized value, p. u.) | CD-F | 44,277.19 | 44,278.55 | 44,278.04 | 44,277.22 |
| | CD-C | N/A | 44,278.63 | 44,278.19 | 44,277.69 |
| | C-F | 44,280.23 | 44,281.59 | 44,281.34 | 44,280.45 |
| | C-C | N/A | 44,280.65 | 44,280.42 | 44,280.39 |
| OF3<br>(normalized value, p. u.) | CD-F | 40,457.35 | 40,459.66 | 40,459.15 | 40,457.39 |
| | CD-C | N/A | 40,459.74 | 40,458.98 | 40,458.40 |
| | C-F | 40,463.21 | 40,464.29 | 40,463.72 | 40,463.24 |
| | C-C | N/A | 40,463.55 | 40,463.54 | 40,463.43 |

On the other hand, it is also observed that (i) a flexible charging rate enables more benefits than a constant charging rate, and (ii) the discharging ability exhibits more profits than charging-only cases in reducing charging costs and flattening daily load.

For more comparison to show the properties of the ISS, a box-plot for CD_C and CD_F mode with the OF1 objective is illustrated in Figure 6. From the distribution of the optimized outcome, it demonstrates that an ISS can achieve more consistent results than that of the other two methods. The robustness and good convergence of this novel algorithm are thus demonstrated.

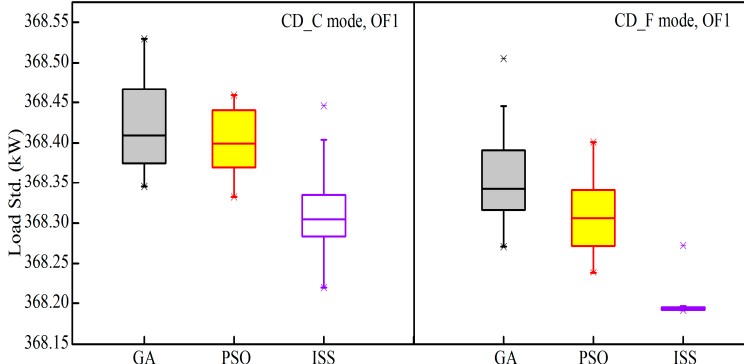

**Figure 6.** Box-plot for CD_C and CD_F mode with objective OF1.

Table 5 gives the average computational time of the respective algorithms for performing 50 simulations. It is clearly shown that the computational time of ISS is significantly shorter than that of the GS, GA, and PSO, which is the main advantage of the proposed algorithm. Furthermore, it can also be observed that the computational time is generally longer for (i) bidirectional charging cases as compared to the charging-only cases, and (ii) a flexible charging rate as compared to constant charging rate.

To examine the computational efficiency of the ISS scheduling algorithm, the iteration behaviors of GA, PSO and ISS, with trial simulations for the three objective functions under CD-F mode and CD-C mode are shown in Figures 7 and 8. The convergence behaviors of C-F mode and C-C mode present similar properties respective to that of the CD-F and CD-C modes, hence their iteration figures are omitted here. As demonstrated in Figures 7 and 8, despite the strong global search ability, the convergence process of both GA and PSO methods are very slow and may easily get trapped into local optima, whereas ISS always converges with far fewer iterations.

In terms of robustness, convergence, and efficiency, the proposed ISS method has adapted very well to single EV charging cases.

**Table 5.** Computational time for a single EV for the methods under four charging modes (In seconds).

| Objective Function | Type | GS | GA | PSO | Proposed ISS |
|---|---|---|---|---|---|
| OF1 | CD-F | 93.16 | 11.67 | 8.51 | 1.18 |
| | CD-C | N/A | 9.38 | 6.24 | 0.98 |
| | C-F | 49.39 | 12.07 | 7.43 | 1.22 |
| | C-C | N/A | 9.97 | 5.98 | 0.86 |
| OF2 | CD-F | 80.39 | 11.13 | 6.88 | 1.14 |
| | CD-C | N/A | 9.77 | 6.15 | 0.95 |
| | C-F | 21.41 | 12.31 | 7.32 | 1.03 |
| | C-C | N/A | 9.62 | 6.47 | 0.81 |
| OF3 | CD-F | 43.61 | 12.65 | 6.42 | 1.25 |
| | CD-C | N/A | 9.38 | 5.97 | 0.94 |
| | C-F | 31.43 | 13.81 | 6.26 | 1.12 |
| | C-C | N/A | 8.88 | 5.83 | 0.83 |

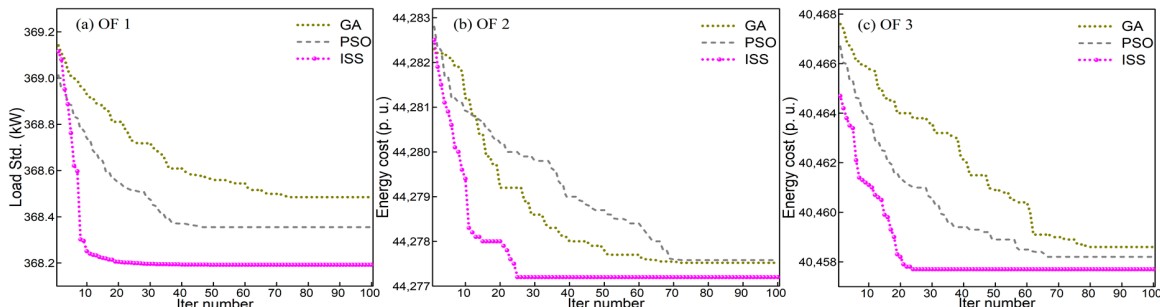

**Figure 7.** Iteration map for single EV with different objectives under CD-F mode. (**Left**) Load standard deviation with OF1; (**Center**) Energy cost with OF2; (**Right**) Energy cost with OF3.

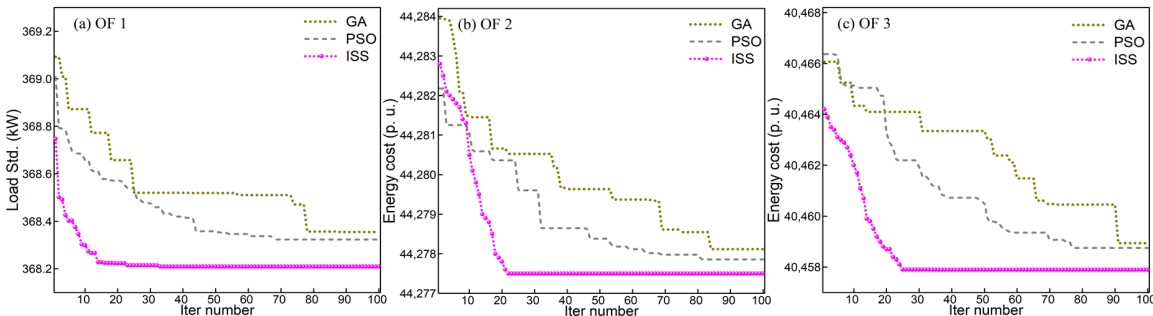

**Figure 8.** Iteration map for single EV with different objectives under CD-C mode. (**Left**) Load standard deviation with OF1; (**Center**) Energy cost with OF2; (**Right**) Energy cost with OF3.

(2)  Group EV charging

For massive amounts of EVs connecting to a charging station or regulated by an aggregator, the proposed GA-ISS hybrid method can help smooth the load profile or minimize the overall charging cost. For illustration purpose, CD-F mode with the aforementioned three objectives is utilized as the testing scenario for the group EV charging case. The desired SOC for all EVs is set to 90%. The maximum running time of the ISS at the initial phase is set as 0.8 s for an individual EV. For comparison and highlighting the benefit of scheduling for massive EVs:

a.   The dumbing control method (DCM), which charges all EVs as soon as they are plugged in, is tested;

b.   A GA-PSO hybrid method from [41] is chosen for comparison, since it can obtain better solutions along with less variation and processing time in comparison to other common heuristic methods. The population size and iteration number for GA-PSO and GA-ISS are set to be 20 and 50, respectively. Moreover, during the inner optimization for EV scheduling of GA-PSO and GA-ISS, the determination of a single EV state via PSO/ISS is set to be stopped when the minimum criteria of the solution quality is satisfied. In Table 6, the detailed parameters of the algorithms utilized for group EV charging are displayed.

c.   A commercial CVX [42] toolbox is also simulated to perform the EV charging scheduling method, since it can simultaneously calculate all the variables and constraints and obtain the global optimized result. This approach is herein named global control.

**Table 6.** Parameters of the algorithms utilized for group EV charging.

| Parameter | DCM | GA-PSO | Global Control | GA-ISS |
|---|---|---|---|---|
| Mutation rate | N/A | 0.2 | N/A | 0.2 |
| Crossover rate | N/A | 0.4 | N/A | 0.4 |
| Population size | N/A | 20 | N/A | 20 |
| Iteration number | N/A | 50 | N/A | 50 |

Figure 9 demonstrates the simulation load profiles of these approaches with different optimization goals, chosen from one simulation result. As can be seen, the dumbing method can amplify the peak, indicating the necessity of EV charging scheduling. On the other hand, it is also shown that the proposed GS-ISS method obtains far better results than that of the GA-PSO method, and it can deliver a performance comparable to global control solved through the CVX solver, which requires a commercial licensing payment.

In addition, the final load profile of TOU price (OF2) is more fluctuated when compared to the LP price case (OF3), as shown in Figure 9b,c. As a matter of fact, the LP price load curve is almost the same as the OF1 one, which is obtained with the aim of load curve flattening. This suggests that an appropriate pricing strategy can help smooth the daily load curve with EV charging/discharging.

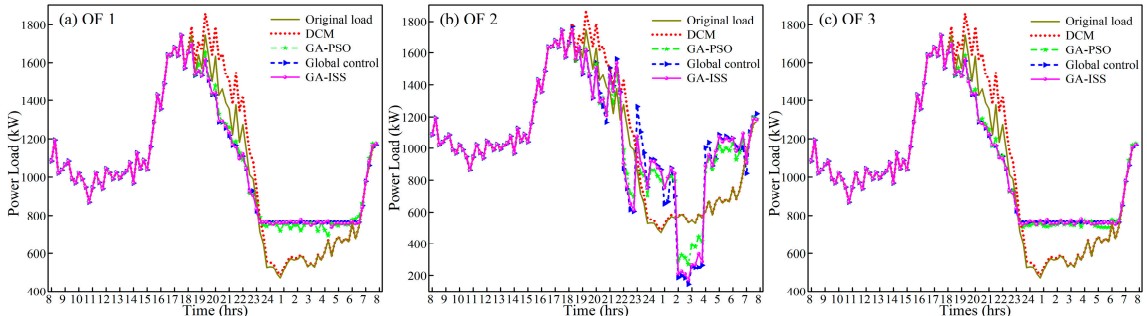

**Figure 9.** Load profiles for group EV charging of DCM, GA-PSO, Global Control, and GA-ISS under CD-F mode.

The average computational time as well as the optimization results for 20 simulations are listed in Table 7. It can be seen that the time spent by the hybrid GA-ISS approach is comparable to that of the global control, but outperforms the GA-PSO method. This verifies the effectiveness of the GA-ISS method for large scale EV scheduling. However, it should also be noted that the CVX solver may become stalled when more EVs are involved (e.g., when the EV number of the fleet changes to be 200, the computer may run out of memory with the tested PC), and hence solution may not be guaranteed for the global control.

**Table 7.** Outcomes of different solving methods.

| Method | OF1 | | OF2 | | OF3 | |
|---|---|---|---|---|---|---|
| | **Result** | **Time** | **Result** | **Time** | **Result** | **Time** |
| GA-PSO | 298.02 | 337.6 | 44,615.2 | 291.2 | 41,274.5 | 359.7 |
| Global Control | 289.39 | 92.6 | 44,538.1 | 87.7 | 41,222.2 | 94.2 |
| GA-ISS | 291.75 | 112.9 | 44,558.5 | 105.9 | 41,229.2 | 117.1 |

## 5. Conclusion and Future Work

This paper presents the detailed derivation of two algorithms: an ISS framework to deal with single EV charging, and a hybrid GA-ISS method comprised of GA theory and the proposed ISS approach for handling a large-scale EV fleet. The main contribution is that the proposed algorithms can support both V2G and G2V, with either a flexible or constant charging power rate. It has been demonstrated that the ISS algorithm is the most computationally efficient way to obtain attractive performances among other methods including GS, GA, and PSO for a single EV charging scenario. On the other hand, for group EV charging, the GA-ISS approach has also been shown to be an extremely effective approach.

The proposed ISS algorithm and GA-ISS hybrid method are shown to be promising in terms of both efficiency and accuracy, thus providing potential techniques for the implementation of EV

charging controls. The proposed algorithms are suitable to be implemented to schedule EV charging in many occasions, such as home charging, EV aggregations (stations), at power companies, etc. It should also be noted that this paper is focused on the issue of EV charging scheduling. Currently, energy storage is increasingly applied in power grids. The charging powers of energy storages are normally much higher than that of EVs, thus can exhibit a greater impact than EVs. Many characteristics of energy storage are similar to EVs and it is valuable to upgrade the algorithm for a wider usage, including the issue of energy storage.

Future work can be performed to incorporate more considerations for EVs, such as the various charging rates of a single EV battery, advanced modeling for SOC impacts on the EV charging rate, EV battery degradations, etc. Studies on the essential coordination between EVs and other power system components (renewable energy resources, home appliances, etc.) can be also addressed to accomplish more comprehensive and practical EV charging algorithms.

**Author Contributions:** T.M. and X.Z. were responsible to the conceptualization of the algorithms; T.M. conducted the investigation, designed the methodology and completed the original draft; X.Z. and B.Z. gave suggestions on the proposed idea and simulation cases; T.M. and X.Z. revised the paper based on comments from the reviewers.

**Funding:** This paper was supported by the project "Study on intelligent simulation technologies and panoramic simulation platform design for power market" of China Southern Power Grid (Project number: ZBKJXM20170082).

**Conflicts of Interest:** The authors declare no conflict of interest.

## Nomenclature

| | |
|---|---|
| ST | Time interval set |
| SE | EV set |
| $i$ | Time index |
| $j$ | EV index |
| $n$ | Total number of timeslots |
| $P_{EV,j}^i$ | Charging power value for EV $j$ in time $i$ |
| $P_{EV,j}^{\min}$ | Lower charging power limit for EV $j$ |
| $P_{EV,j}^{\max}$ | Upper charging power limit for EV $j$ |
| $Soc_{j,i}$ | SOC for EV $j$ in time $i$ |
| $Soc_{j,start}$ | SOC value at the beginning for EV $j$ |
| $Soc_{j,\mathrm{end}}$ | SOC value at the end for EV $j$ |
| $Soc_{j,\min}$ | Minimum allowed SOC value for EV $j$ |
| $Soc_{j,\max}$ | Maximum allowed SOC value for EV $j$ |
| $E_{\exp,j}$ | Expectation level for the SOC of EV $j$ |
| $E_{RC,j}$ | Nominal battery SOC of EV $j$ |
| $P_{total}^i$ | Total load demand in time $i$ |
| $P_{ld}^i$ | Active base load in time $i$ |
| $P_g^i$ | Maximum supply power in time $i$ |
| $C_e^i$ | Electricity tariff in time $i$ |
| $C_{TOU}^i$ | TOU price at time interval $i$ |
| $C_{LP}^i$ | Linear price rate at time interval $i$ |
| $\psi$ | Linear term of linear price |
| $\gamma$ | Constant term of linear price |
| $C_{EV,j}$ | Nominal capacity of EV $j$ |

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
