# Peer review of "Intelligent Energy Management Algorithms for EV-charging Scheduling with Consideration of Multiple EV Charging Modes"

_energies, doi:10.3390/en12020265_

Round 1

Reviewer 1 Report

The authors have done a good job, and they have improved the paper a lot.

The paper clearly reflects the research carried out.

The paper presents a detailed design of ISS and compared with methods based on GS (Global Search), GA (Genetic Algorithm)  and PSO (Particle Swarm Optimization), the outcome verified ISS can produce attractive results with  significantly short computational time. Moreover, to handle large scale EV charging scenario, a hybrid method comprised with GA and ISS approach has been further developed. Simulation results also verified its prominent performance plus superb low computational time.

Author Response

Reply: The authors are grateful to your review and valuable comments. Thanks a lot!

Reviewer 2 Report

The article is interesting, well structured and organized.

Introduction:

It is very convenient to make a more updated bibliographic review, there are several articles of interest as: 

Alonso, M., Amaris, H., Germain, J. G., & Galan, J. M. (2014). Optimal charging scheduling of electric vehicles in smart grids by heuristic algorithms. Energies, 7(4), 2449-2475. doi:10.3390/en7042449

Wang, B., & Yang, J. (2018). Optimal electric vehicle charging scheduling with time-varying profits. Paper presented at the 2018 52nd Annual Conference on Information Sciences and Systems, CISS 2018, 1-6. doi:10.1109/CISS.2018.836228

García-Álvarez, J., González, M. A., & Vela, C. R. (2018). Metaheuristics for solving a real-world electric vehicle charging scheduling problem. Applied Soft Computing Journal, 65, 292-306. doi:10.1016/j.asoc.2018.01.010

Simulation Results:

In the comparison of the different Group EV charging it is necessary to describe the crossver and mutation values applied to the different algorithms. It is necessary to show a comparison between the values because with the implementation of ISS, a low mutation percentage would harm the other algorithms against GA-ISS.

Also when it want to compare different genetic algorithms it is necessary to compare for different population sizes and iterations, in order to demonstrate the convergence of different solutions. Although if we really want the speed we should replace the number of iterations to stop by another parameter such as "the highest ranking solution's fitness is reaching" or "a solution is found that satisfies minimum criteria"

Conclusion:

It cannot be concluded that one method is better than another when a sufficiently deep comparison has not been made, either with the proposed problem or in the face of well-known mathematical problems where the efficiency of the new algorithm can really be seen.

Author Response

Reply: The authors would like to thank you for your sincere comment and the paper has been refined based on your comment. Please find the detailed response in the attached PDF file. Thanks a lot!

Reviewer 3 Report

1. It would be good to consider EVSE with paired energy storage. This configuration will become increasingly common as DCFC units increase in charge power to extreme fast charging (XFC) rates of 400 kW, in order to avoid exorbitant demand charges
2. Line 90: It's unclear what is meant by "license concern". This should be explained.
3. Line 91: Can the CVX, GAMS, and CPLEX algorithms not solve more complicated problems? This claim requires evidence.
4. Line 96: Does the "longer computational time" preclude the usage of these methods for real-time charging scheduling? Do the results from any other algorithm demonstrate that real-time scheduling is possible?
5. Line 105: The charging rate is determined by the EV, not the EVSE unit (the EVSE unit merely "indicates" to the EV what the maximum available power is). The rate will change as the SOC changes as well. How will this impact the method? At the very least, this should be mentioned in the text.
6. Line 106: Sentence beginning with "Whilst" is an incomplete sentence.
7. Line 116: "population-based"
8. Section 3.1 Assumptions: Has the assumption been made that the EVs all have the same energy capacity in their ESS? This needs to be described explicitly. Table 2: Ok, it's clear that 24 kWh has been assumed. This energy capacity is from the first generation of EVs and should be increased. 60 kWh is common for the the Chevrolet Bolt and Tesla 3.
9. Table 3: It appears that the maximum possible charge rate is 10 kW. It would be good to include DCFCs, but this should at least be mentioned in the "future work" section.
10. Conclusion section: A discussion of what are the real-world implications of the work is necessary, including speculation (at least) on how the modeling/simulation can be validated. Further, describing what is the next step in the research is very important.

Author Response

Reply: The authors thank you for your constructive comments. Based on your comments, the paper has been refined with highlighted texts. Please find the detailed response in the attached PDF file. Thanks a lot!

Round 2

Reviewer 2 Report

Accept in present form